# Research

ecology

altitude, climate change, land use change, plant–pollinator, wild bees, conservation

**Author for correspondence:**
Leon Marshall
e-mail: leon.marshall@ulb.ac.be

# Bumblebees moving up: shifts in elevation ranges in the Pyrenees over 115 years

Leon Marshall[1,2], Floor Perdijk[1], Nicolas Dendoncker[3], William Kunin[4], Stuart Roberts[5] and Jacobus C. Biesmeijer[1,6]

[1]Naturalis Biodiversity Center, PO Box 9517, 2300 RA Leiden, The Netherlands
[2]Agroecology Lab, Université libre de Bruxelles (ULB), Boulevard du Triomphe CP 264/2, 1050 Brussels, Belgium
[3]Department of Geography, Institute of Life, Earth and Environment (ILEE), University of Namur, Rue de Bruxelles 61, 5000 Namur, Belgium
[4]School of Biology, University of Leeds, Leeds LS2 9JT, UK
[5]Centre for Agri-Environmental Research, School of Agriculture, Policy and Development, University of Reading, Reading RG6 6AR, UK
[6]Institute of Environmental Sciences (CML), Leiden University, Einsteinweg 2, 2333 CC Leiden, The Netherlands

(iD) LM, 0000-0002-7819-7005

In a warming climate, species are expected to shift their geographical ranges to higher elevations and latitudes, and if interacting species shift at different rates, networks may be disrupted. To quantify the effects of ongoing climate change, repeating historical biodiversity surveys is necessary. In this study, we compare the distribution of a plant–pollinator community between two surveys 115 years apart (1889 and 2005–06), reporting distribution patterns and changes observed for bumblebee species and bumblebee-visited plants in the Gavarnie-Gèdre commune in the Pyrenees, located in southwest Europe at the French–Spanish border. The region has warmed significantly over this period, alongside shifts in agricultural land use and forest. The composition of the bumblebee community shows relative stability, but we observed clear shifts to higher elevations for bumblebees (averaging 129 m) and plants (229 m) and provide preliminary evidence that some bumblebee species shift with the plants they visit. We also observe that some species have been able to occupy the same climate range in both periods by shifting elevation range. The results suggest the need for long-term monitoring to determine the role and impact of the different drivers of global change, especially in montane habitats where the impacts of climate changes are anticipated to be more extreme.

## 1. Introduction

Climate change is expected to increase temperatures globally, particularly at high elevations and latitudes [1]. Climate change impacts the spatial distribution of biodiversity [2], often driving species to higher elevations and latitudes [3–5]. This leads to an increase in species richness at cooler latitudes and elevations and may result in species from these cooler areas 'falling off the top of the mountain' as suitable conditions no longer exist or as species, which dominate in warmer areas, outcompete them [6]. Shifts in elevation are expected to be more apparent and quicker than shifts in latitude [4]. Areas of high elevation often contain rapidly changing climate conditions across short distances and therefore are easier for species to follow [7]. Overall, the expected patterns of range change at high elevations include the extinction of populations at lower elevation and more species colonizing higher elevations. However, in practice, species from lower elevations may not adequately counteract the loss of high elevation species going extinct or shifting

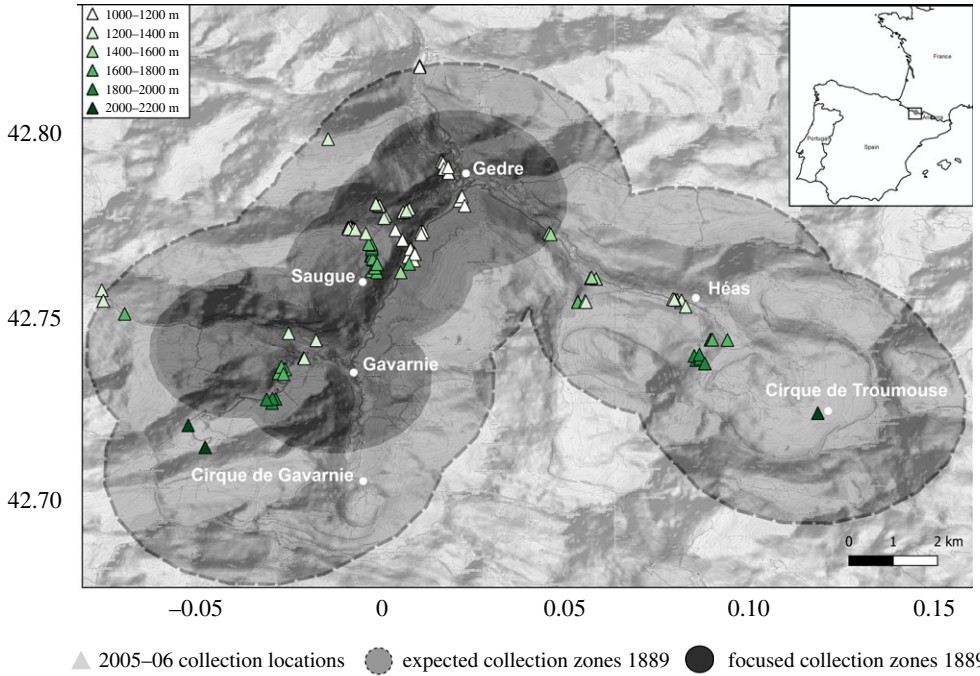

**Figure 1.** Locations of survey sites in 1889 and 200–06 in the Pyrenees National Park. Grey, transparent circles represent the estimated sampling locations in 1889 based on all locations mentioned by MacLeod. Black, transparent circles represent the areas where MacLeod conducted most of his surveys (greater than 85% of collections). Green triangles represent exact sampling locations in 2005–06. Inset map shows location in Europe (field site located within black square). Sites from 2005–06 are coloured according to elevation, light (low elevations) to dark (high elevations). (Online version in colour.)

even higher, and this may result in the dominance of widespread species at all elevations [8]. These shifts are likely to have causal effects on interacting species. Where interacting species respond differently to warming temperatures or differ in dispersal ability, the differences in the direction or speed of range shifts could lead to phenological or spatial mismatches in plant–pollinator relationships [9]. Alternatively, interactions with strong interdependencies may constrain a species' ability to respond to changing conditions [10].

Elevation gradients in alpine habitat provide *in situ* opportunities to study how species adapt to changing environments [11]. Bumblebees are an ideal group to illustrate distribution patterns at high elevations. Many bumblebee species are adapted to low temperatures [12]. As temperatures increase these adaptations will be less essential and may lower survival rates at previously suitable elevations [13]. Repeating historical surveys is key to understanding the changing ranges of high-elevation species [14]. Most studies comparing historical and modern surveys of insect distributions in high elevation areas show species increasing in elevation. Over 42-years, moths on Mount Kinabalu, Borneo, shifted in elevation by an average 67 m [15]. Over 35-years in the Sierra Nevada Mountains, most butterfly species shifted significantly higher in mean elevation, consistent with the climate warming in the area [16]. By contrast, in northern Sweden two surveys 60 years apart did not show a clear trend of insect species moving to higher elevations [17]. Bumblebees are unlikely to be able to freely track climate changes, as they rely on plant species as food sources. Plant species have also shown significant increase in elevation in several areas. In western Europe comparing plant species' mean elevation over the last 100 years shows them shifting on average by 29 m per decade [18]. Due to the reliance of bumblebees on plants, one would expect that this interaction will also shift with climate change. However, loss of spatial occurrence as well as phenological shifts may lead to

mismatches in co-occurrence of bees and plants, as observed over a 120-year timespan in Illinois [19].

In this study, we compare a high elevation community of bumblebees, and insect-visited plants in the Pyrenees 115 years apart. Specifically, we used observations of plant and plant–visitor communities surveyed in 1889 [20] and compared them to surveys conducted in the same areas in 2005–06. We aimed to test if the bumblebee and plant community had altered in composition and/or mean elevation over 115 years; and whether bumblebee visitors had shifted with the plants they visit.

## 2. Methods

### (a) Study area
We studied the long-term temporal changes of bumblebees and the plants they visit in the area of the commune Gavarnie-Gèdre in the Hautes-Pyrénées department of France, located next to the border with Spain (figure 1). The surveyed area is part of the Pyrenees National Park (est. 1967) located in the western part of the Pyrenees. The elevation in the national park ranges from approximately 1000 m.a.s.l to its maximum of 3298 m.a.s.l., the Vignemale Peak. The region straddles the borders of the Atlantic and Mediterranean biogeographic zones and therefore is home to a broad and diverse flora and fauna with many endemic species [21]. While protected, the region is still home to settlements and agricultural land, with settlements usually at lower elevations and agricultural areas higher up the mountainside, up to 2000 m [22]. Broadly, the vegetation of the region is described as hay meadows and pine forest, with the tree line around 2000 m [23]. The area has both oceanic and montane climates with an average annual temperature of approximately 6.5°C and average annual rainfall of 1049 mm.

### (b) Climate change and land use changes
We generated climate data using the software package ClimateEU [24] (v. 4.63). These climate measures are modelled climate

changes due to the absence of spatially and temporally accurate data for the study site from the nineteenth century. The dataset provides minimum, maximum and mean temperature records for sample locations with known elevation. We extracted these metrics at a $1 \times 1$ km grid resolution for a 10 km buffer surrounding the centroid of all collection records. Therefore, the climate changes measured are representative of the entire study area and not only sites where species where collected. We aggregated the values for each decade by taking the mean value across all 10 years. Since the climate records available in the ClimateEU software start at 1900, we took the decade 1900 to 1910 as a proxy for the period 1885 to 1895. For the modern-day records, we also aggregated the data between 2000 and 2010 to the mean value of each metric across all 10 years. We then compared each of the temperature metrics using paired two sample $t$-tests to examine whether temperature values were significantly different between the two periods, both annually and for August. We also calculated the annual mean temperature and the mean temperature of August for all years between 1900 and 2006 to test whether there was a significant trend in changing temperature. For each bumblebee species observed in both periods, we calculated the difference in the average mean temperature of August for the occupied range. We used the non-parametric, Kruskal–Wallis one-way analysis of variance test to see whether the climate range occupied was warmer in 2005–06 than 1889 for each species.

Land use/land cover (here after referred to as land use) maps for the Pyrenees in 1889 were not available. To estimate land use change in the area, we used historic reconstruction maps for Europe [25]. These maps represent modelled reconstructions of land use in Europe over the twentieth century using a combination of historical land use data sources and a modelling approach called Historic Land Dynamics Assessment or HILDA [26]. This resource provides a rough estimate of how the land use in the study area has shifted in the past century at a $1 \times 1$ km grid resolution. We used land use maps from two decades to show changes in the periods of the surveys (1910 and 2010). Due to the coarseness of the land use data, there were not enough grid cells to analyse the change in land use in each elevation surveyed. Therefore, we split the mountain into three regions of different elevation, 1000–1400 m, 1400–1800 m and 1800–2200 m. We extracted the land use within each elevation profile at a $1 \times 1$ km grid resolution for a 10 km buffer surrounding the centroid of all collection records within the same elevation profile. Therefore, the land use changes measured are representative of the entire study area and not only sites where species were collected. $\chi^2$ tests were used to assess differences in the proportions of each land use class at each elevation zone between the time-periods. The land use classes available in the historic land use maps include forests, grasslands, cultivated land, human settlements, water and other. The 'other' category includes areas of ruderal vegetation, bare soil, rocks and other parts of the landscape difficult to classify.

## (c) Bumblebee and plant surveys

We focused the study on wild bumblebees, as they are abundant in the historical survey, are adapted to high elevation regions, well known, easy to survey, and able to be identified to species level. Two separate surveys in the region, both conducted in August, were compared. Between 5–31 August 1889, biologist and naturalist Prof. Julius MacLeod sampled the plant and plant visitor communities, in seven locations across the area (figure 1). In August 2005 and 2006, efforts were made to resample the same areas as MacLeod had visited, this time focussing the survey on the plant species with insect visitors recorded in 1889.

## (d) 1889 Collections

Julius MacLeod sampled plant communities in the areas of Gèdre (1000 m), Cascade de Gavarnie (1500 m), port de Gavarnie (2300 m), cirque de Troumouse (2000 m), the brêche de Roland (2800 m), Saugué (1500–1650 m) and Héas (1450 m). These areas thus encompass elevations ranging from 1000 to 2800 m.a.s.l., plant visitors were recorded from 1000 to 2100 m. He published an account of the plants and plant visitors he observed in 1891 in 'De pyreneënbloemen' [20]. The names of these seven areas and the exact elevation at which he made observations was all the information published regarding his sampling locations. See electronic supplementary material, appendix A for a full description of each area sampled. The goal of MacLeod's survey was to make a comparison of the floral community along habitat and elevation gradients in the Pyrenees. Nonetheless, MacLeod collected and identified all insect visitors observed when surveying the plant community. MacLeod surveyed 263 separate plant species with 569 separate insect visitors. The bumblebees collected by MacLeod were identified by Prof. Otto Schmiedeknecht. A number of species names did not correspond with present-day terminology and we used Schmiedeknecht's publication, Die Hymenopteren Mitteleuropas [27], to compare with the checklist of bumblebees from the Natural History Museum, London [28], to determine the correct taxonomic names which would correspond to present-day bumblebee nomenclature. The plant species, identified by MacLeod himself, were compared using the 'The Plant List', an online resource with historical synonyms of many global plants (http://www.theplantlist.org/).

## (e) 2005/06 Collections

In 2005 (8–25 August) and 2006 (14–31 August), two surveys were conducted to analyse the plant visitor community of the most visited plant species in 1889, in the same areas that MacLeod sampled. The sampling locations were estimated based on the seven sampling locations listed above and the exact elevations reported by MacLeod for each plant species and insect visitor. In 2005–06, the target was the plant visitors, and a selection of plants was made first based on MacLeod's findings, i.e. observing the same plant species that were observed by MacLeod. However, to maximize sampling of the pollinator community, additional observations were made on other well-visited abundant flowering plant species. In summary, the first goal was to find a match with the location and plant species observed by MacLeod. If that was not possible, an alternative nearby location at similar elevation with the same plant species observed by MacLeod was searched for. In addition, visitors of other flowering plants at the original MacLeod locations were observed and lastly, other flowering plant species were observed at locations corresponding in elevation, but different from the MacLeod locations. In this way, we strived to make the 1889 and 2005–06 studies as comparable as possible in terms of visited plants, locations and elevations. At each location surrounding, the area mentioned by MacLeod for each of the plant species chosen a plot was observed for 15 min. Plot size was determined as the largest area of a patch of flowers on which all visitors could be observed (from a few to about a dozen square metres depending on the growth and blooming of the plant species). During the 15 min observation window all bumblebee visitors were caught and later identified by experts. The only exception was *B. gerstaeckeri* which was identified by sight because it is unmistakable in the field and rare. The surface area of each plot was measured, and its flower density was recorded. The elevation and GPS coordinates (WGS 84) for each plot were also recorded.

## (f) Community change

We used rarefaction methods to account for the differences in sampling intensity between the two periods. Using the iNEXT package [29,30] (v. 2.0.19) in R we calculated interpolated and

extrapolated estimates of species richness (Hill number 0) for three elevation ranges at different sampling intensities.

## (g) Elevation change

We examined elevation change for all bumblebees and plants observed at least twice in both periods. MacLeod's descriptions of his sampling locations are not explicit enough to attribute exact point locations to the collection records. Therefore, as we do not know exactly where MacLeod sampled, we grouped the occurrences into elevation ranges rather than sites (MacLeod provided elevation values for each observation). We split the occurrences into elevation ranges of 200 m, i.e. from 1000–1200 m to 2000–2200 m. The number of occurrences collected at each elevation range are not uniform between the two periods. Therefore, to avoid any bias of oversampling at certain elevations, we used the approach of Chen et al. [15] to determine the average elevation of individuals in 1889 and in 2005–06. We follow Chen's recommendation and at each elevation range the period with the greater number of records was resampled to coincide with the period with lower sampling intensity and then the mean elevation of each species was calculated. This was repeated 1000 times. The difference between periods per species was calculated in each randomization and the average difference was used. We then compared the elevation ranges in both periods to assess whether there was an increase or decrease in elevation overall and per species using $t$-tests or a Wilcoxon rank-sum test when the sample means were not normally distributed.

To test if the elevation shift in bumblebees was comparable to shifts in the plants they visit, we calculated the mean elevation shift of the plants visited for each species. To reflect floral preferences between species, we calculated a weighted mean, weighted by the frequency of observations on each plant. We then tested using a linear model whether there was a positive correlation between the elevation shift of bumblebees and the elevation shift of the plants they visit in the region. Furthermore, to examine specific relationships between bumblebees and the plants they pollinate we selected the three most visited plants across the two periods and plotted their elevation profiles. Alongside the plants' elevation we show how bumblebee visitation proportions at different elevations have changed between the two periods.

## 3. Results

### (a) Climate and land use changes

At the landscape level, we observed considerable modelled climate change between the two periods. The mean annual temperature significantly increased by 0.02°C per year ($f = 2667$, d.f. = 38802, $p \leq 0.001$, electronic supplementary material, figure S1), equivalent to a 2.3°C increase over 115 years. Furthermore, the average mean, minimum and maximum temperatures of August between 1901–1910 and 2001–2010 showed significant differences. The mean temperature increased on average by 2.1°C ($t = 1351.7$, d.f. = 355, $p \leq 0.001$). The minimum temperature increased on average by 2.3°C ($t = 1210.1$, d.f. = 355, $p \leq 0.001$) and the maximum temperature by 1.9°C ($t = 990.9$, d.f. = 355, $p \leq 0.001$). The equivalent average temperature in August is now found on average 425 ± 44 m higher, the minimum 513 ± 24 m higher and the maximum 299 ± 80 m higher (electronic supplementary material, figure S2a).

Most bumblebee species showed a lower increase in their occupied elevation ranges of the average August temperature than the climate change modelled for the region (2.1°C). The average mean difference in August temperatures of the

occupied ranges of all species was 1.3°C. *Bombus soroeensis* showed the greatest difference, occupying a climate range 3.4°C warmer on average. The smallest change was observed for *B. gerstaeckeri*, 0.24°C. Five species show no statistical difference ($p \geq 0.05$) between their climate ranges in both periods. Including, *B. gerstaeckeri*, *B. wurflenii*, *B. ruderarius*, *B. mesomelas* and *B. lapidarius* (figure 2).

We saw a statistically clear change in the land-cover of different elevation zones between 1910 and 2010 (electronic supplementary material, figures S2b and S3). In 1910, elevations between 1000 and 1400 m were covered 22% by forest, 67% by grassland and 11% by other; by 2010 this has changed to 67% forest, 33% grassland and 0% other (chi-square: $\chi^2 = 8$, d.f. = 2, $p = 0.02$;). At 1400–1800 m elevation, similar changes occurred, with forest increasing from 15% to 30% and grassland decreasing from 74% to 44%, and other land shifted from 11% to 25% (chi-square: $\chi^2 = 9.9$, d.f. = 2, $p = 0.007$). At 1800–2200 m land use changed less than at lower elevation. Forest was only a small proportion and increased from 1% to 9%, grassland stayed the same at 51% and other land changed from 48% to 40% (chi-square: $\chi^2 = 7$, d.f. = 2, $p = 0.03$).

### (b) Community composition

The lower sampling intensity displayed in the extrapolated rarefaction curves suggested that there was potentially a greater diversity in bumblebee species in 1889. The confidence intervals, however, provide no evidence that the diversity of bumblebees was significantly different between periods (electronic supplementary material, figure S4). The total number of bumblebee species found in 1889 was 16 and increased to 17 in 2005–06. Twelve species were found in both surveys. Unique to 1889 were *B. mendax*, *B. monticola*, *B. mucidus*, and *B. pratorum*. *Bombus mendax* and *B. mucidus* were singletons, but *B. monticola* (2% proportion of all bumblebees collected in that period) was found more abundantly. Singletons found in 2005–06 but not in 1889 include *B. rupestris*, *B. sylvarum*, and *B. sylvestris*. *Bombus bohemicus* (5%) and *B. pyrenaeus* (2%) on the other hand were absent in 1889 and found with a moderately high abundance in 2005–06

### (c) Elevation shifts

We measured the change in elevation of the 12 bumblebee species present in both periods (figure 3a). The mean elevation shift was calculated based on random resampling of the data at different elevation profiles to match the period with lower sampling intensity (see methods and Chen et al. [15]). The lowest elevation surveyed was 1000 m. We observed a statistically clear overall shift upwards of 129 m for all bumblebee species ($t = 2.5$, d.f. = 11, $p = 0.039$, 95% CI = 14.3, 243.81). The species that shifted the most in elevation were *B. wurflenii*, *B. lapidarius*, and *B. gerstaeckeri*. *Bombus wurflenii* had an average elevation shift of 326 m with a minimum elevation of 1000 m and maximum of 1700 m in 1889 and a minimum elevation of 1491 m and maximum of 2200 m in 2005–06 (figure 3a). *Bombus lapidarius* shifted on average 360 m. *Bombus gerstaeckeri* had an average elevation shift of 400 m with a minimum elevation of 1600 m and maximum of 1900 m in 1889 and a minimum of 2100 m and maximum of 2150 m in 2005–06. The only species that showed a large downhill trajectory between the periods is *B. soroeensis*, which had an average elevation decrease of 209 m, however, *B. soroeensis* expanded its range

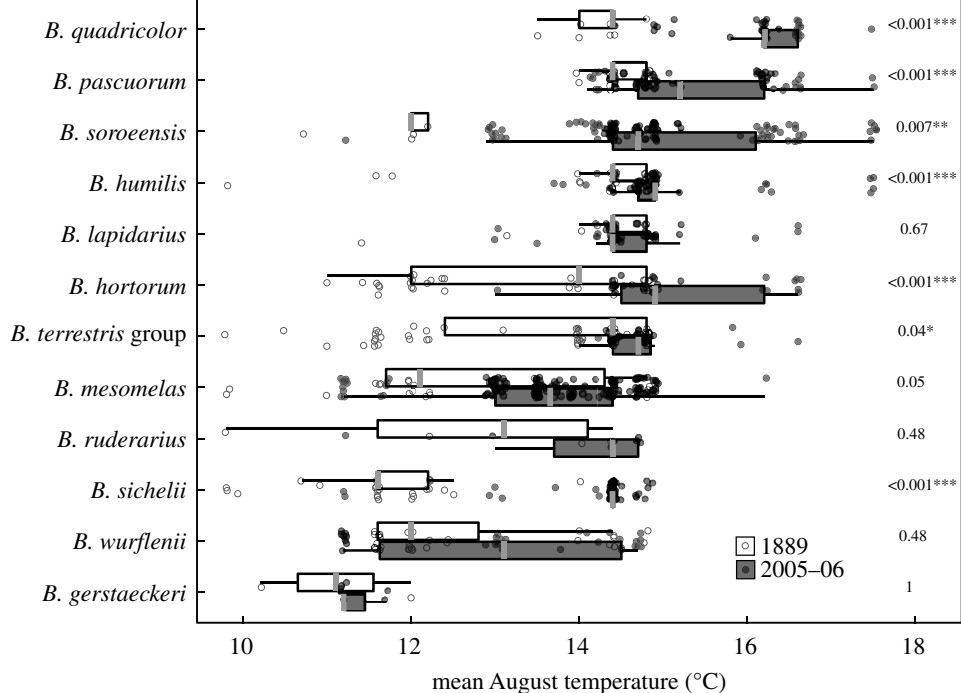

**Figure 2.** Box and whisker plot showing the climate ranges occupied by all bumblebee species observed in both periods. Grey bar represents the median climate value for that species in that period. The box shows the 25th to 75th percentile range of climate values. The lines extend to the median ± 1.58 * IQR/sqrt(n) with values to the left or right of these lines considered to be outliers. All recorded observations are shown as open circles for 1889 and grey circles for 2005–06. For each species, we used a non-parametric, Kruskal–Wallis one-way analysis of variance test to see if the climate range occupied in 2005–06 was warmer than 1889, exact $p$-values are shown. *$p < 0.05$, **$p < 0.01$, ***$p < 0.001$.

overall and was found both higher (2200 m) and lower (1000 m) in 2005–06 than in 1889 (1100–1700 m).

We limited our analysis to the plants which were observed in both periods (figure 3b). Of these 26 plant species, 15 were recorded with bumblebee visitors more than once. Species shifted on average 229 m uphill comparing 1889–2005–06 ($W = 55.5$, d.f. = 14, $p = 0.02$, 95%CI = 133.61, 324.71). *Cirsium arvense* (518 m), species of the genus Aconitum (477 m) and *Allium lusitanicum* (466 m) showed the greatest average shift in mean elevation. The relationship between bumblebee elevation shifts and the plants they visit in both periods showed a positive trend. Overall, we see that the bumblebee-visited plants shifted more than their associated bumblebee visitors (figure 3b). The shift in bumblebees and the plants they visited in either period was weakly positively correlated (linear regression: $R^2 = 0.24$, $\beta = 0.25$, $p$-value = 0.1; figure 3c,d). Three species show a negative relationship, *B. sichelii*, *B. hortorum* and *B. soroensis*. These species decreased in overall elevation while the plant species they visited shifted between 200 and 350 m uphill. The positive relationship observed is also dependent on the relationships between *B. wurflenii*, *B. gerstaeckeri* and Aconitum spp. with all three showing shifts over 400 m. Similar shifts in elevation of bumblebees and the plants they visited were observed for *B. pascuorum*, and *B. mesomelas*. Overall, plant species visited shifted further than all bumblebee species except *B. lapidarius*.

The three plants most visited by bumblebees across the two periods were Aconitum spp., *Cirsium eriophorum* and *Carduus defloratus*. These species have shifted 465, 227 and 327 m uphill. Each of the three species were visited by a broad selection of bumblebee species in both periods (figure 4). In 2005–06, all three were still visited consistently by bumblebees at higher elevations. Across the two periods,

15 different bumblebee species were observed visiting these plants (figure 4). For Aconitum spp. the patterns of visitation at higher elevation seen in 2005–06 reflect the species (*B. mesomelas*, *B. wurflenii* and *B. gerstaeckeri*) and proportions observed at lower elevations in 1889 (figure 4b,c) but with a larger diversity of visitors (six) at the higher elevations in 2005–06. For *C. defloratus*, we see a greater diversity of visitors at the lower elevations in 1889 and that *B. terrestris* group is a common visitor in 1889 and was not observed visiting at all in 2005–06. *Bombus mesomelas* is a common visitor of all three species across the elevation range in 2005–06. *Cirsium eriophorum* was also visited by a greater diversity of bumblebee species in 1889 compared to 2005–06.

## 4. Discussion

In this study, we surveyed plant–bumblebee communities in the same area 115 years apart. We observed statistically clear shifts in elevation for bumblebees. We observed similar, slightly larger shifts in the plants visited in both periods. Historical climate and land use changes have occurred in the region. Along with the shift in floral resources, all three are potential drivers of the community patterns observed. The conclusions are restricted by the lower sampling intensity in 1889 but still provide an unprecedented insight into over 100-years of change in a bumblebee, plant community.

The bumblebee communities sampled demonstrated a stable species richness over 115 years, with only slight changes in species composition and proportions. The most likely explanation for the absence of species in either of the periods is sampling intensity and species phenology. The species absent from the present were only observed in very low densities and have been recorded nearby in the region since the

Proc. R. Soc. B 287: 20202201

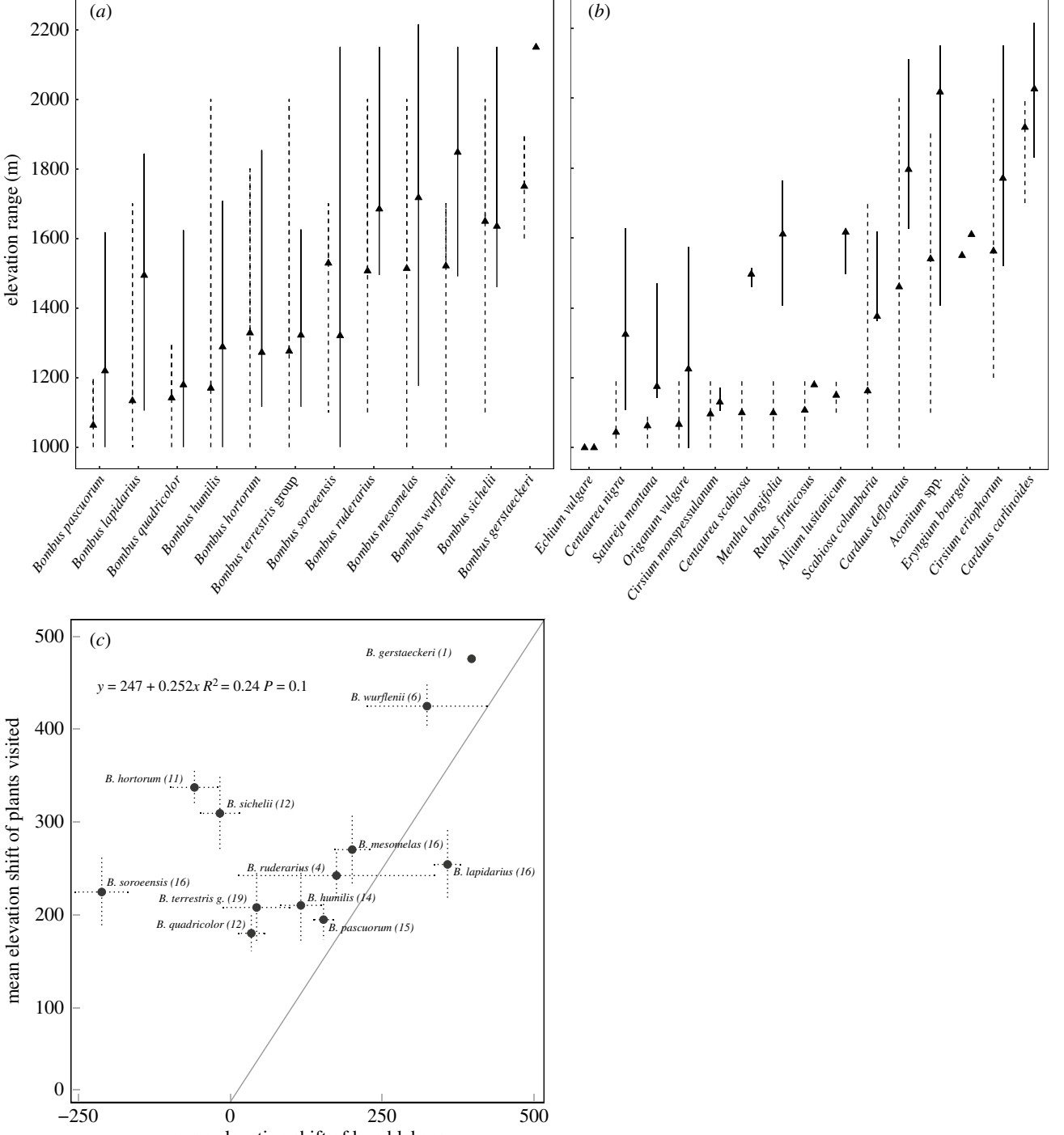

**Figure 3.** Elevation change between 1889 and 2005–06 for (*a*) bumblebees, (*b*) plants visited in both periods in at least two locations. Dashed line: 1889. Solid line: 2005–06. Lines show the minimum to maximum elevation range where the species was observed. (*c*) The relationship between the mean elevation shift in bumblebees and the weighted mean elevation shift of the plants they visited. The mean shift in plants is weighted by the frequency of observations of that interaction. Numbers in brackets refers to the number of plant species used to calculate the shift in visited plants. Dotted lines around each point indicates the standard deviations of the resampled mean elevations. Grey line indicates the line of 1 to 1 ratio, points along this line indicate where the shifts of the plants and the bumblebees would be equal. The equation for the linear model fitted for all points is shown. For all graphs, the mean values are calculated based on resampling the dataset to the lower intensity at different elevation ranges (1000–1200, 1200–1400, 1400, 1600, 1600–1800, 1800–2000, 2000–2200). In each elevation class, the period with the greater number of records was resampled to coincide with the period with lower sampling intensity and then the mean elevation of each species was calculated. *Bombus terrestris* group (or g.) refers to *B. terrestris* and *B. lucorum*, which could not be reliably separated.

2000s [31]. The bumblebees showed an average upward shift of approximately 130 m, in line with similar studies of long-term elevation change of butterflies in other locations [8,15,32]. This study is one of the few studies to show a long-term elevation shift in bumblebees. Franzen & Ockinger [17] measured changes in bumblebee elevation in Sweden but

observed no significant increase across 60 years. At a wider spatial scale Kerr *et al.* [33] observed that in the USA and Europe, southern species showed an overall increase in elevation of approximately 300 m since 1974. This effect varied by species, but the geographical effect of north versus south was significant, with species in the north showing less

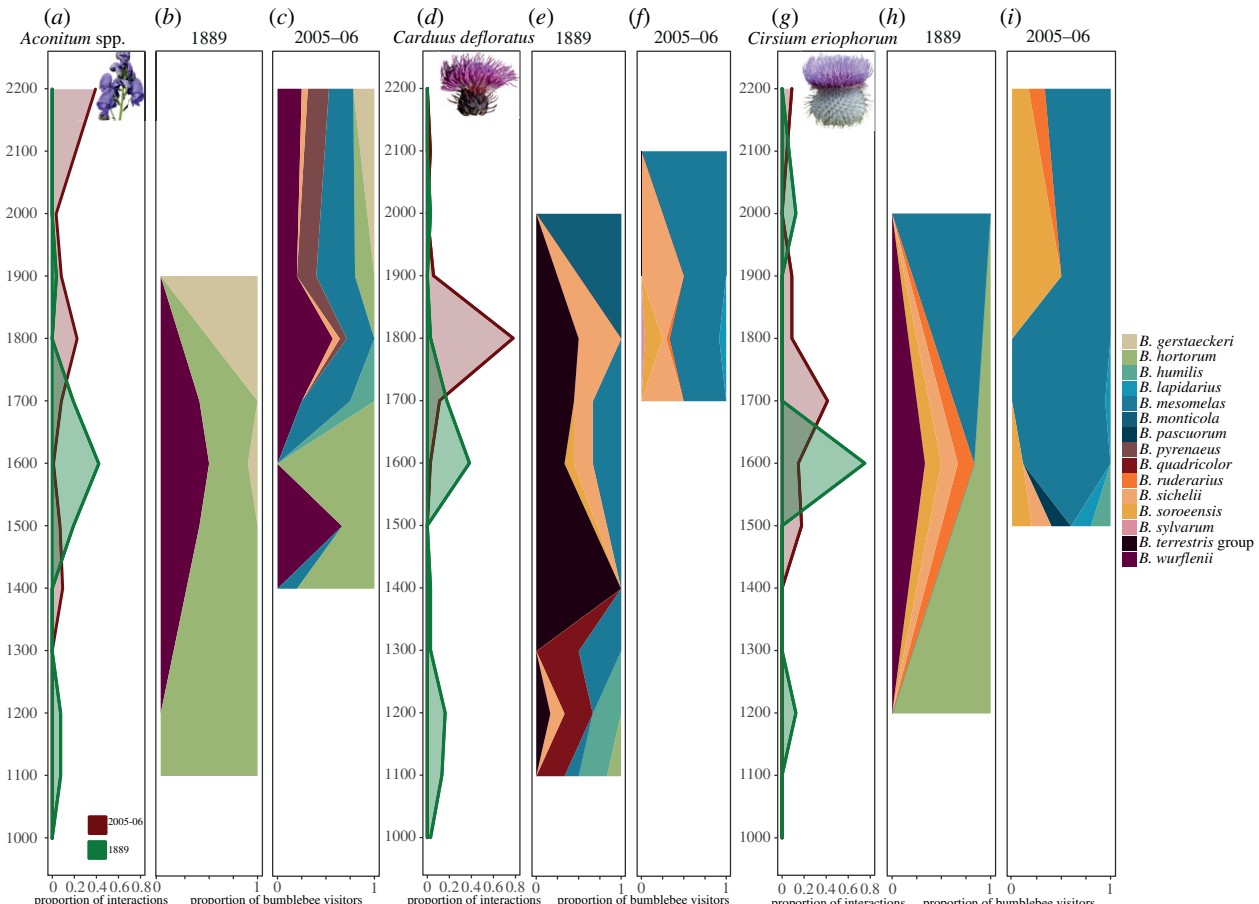

**Figure 4.** Visitation profiles of the three most visited plants across the two periods. Figures (*a*), (*d*) and (*g*) show the visitation frequency along the elevation profile for each period. Figures (*b,c,e,f,h,i*) show the proportion of visits shared between the different bumblebee species at each elevation for the two survey periods (August 1889 and August 2005–06). To aid visualization bumblebee colours are shown continuously across the elevation profile but must be compared with the peaks of the total elevation profiles (*a*, *d* and *g*) to understand the visitation frequency each species is responsible for, as at certain elevations there was no observed visitation. *Bombus terrestris* group refers to *B. terrestris* and *B. lucorum* which could not be reliably separated.

increase in elevation [33]. This supports the observed differences between this study and that by Franzen & Ockinger [17] as the Pyrenees are much further south in Europe than the Swedish mountains. This contrasts with Fourcade *et al.* [34] who observed elevation increases and species richness changes of the bumblebee community in a Norwegian sub-alpine habitat since the 1960s. In the Rocky Mountains in Colorado, USA, Pyke *et al.* [35] found similar results when measuring the elevation increase of bumblebees between 1974 and 2007. As with our study, they found that elevation increase was not consistent for all species.

Most bumblebee species can forage over large distances and while they can occur at very high elevations, they are generally not restricted to specific elevations across their whole range [12,36]. However, in lower, warmer latitudes of the Northern Hemisphere bumblebees can often be restricted to high elevation mountain habitats [37–39]. This suggests some of the bumblebees recorded in this study represent separate high elevation populations while others are part of larger populations ranging from low to high elevations. *Bombus soroeensis* for example seems to have become commoner across a wider elevation gradient over the last century. The results support the hypothesis that all three drivers (climate, land use change and floral resource availability), and potential interactions between them, may have been responsible for the observed changes. Fourcade *et al.* [34] found evidence that

both climate and land use may have been responsible for changes to a sub-alpine bumblebee community but they too were unable to test this interaction directly. Without finer scale historical data, it is difficult to separate the effects of the different drivers. Equivalent minimum, mean and maximum temperatures have shifted between 300 m and 500 m uphill, forest cover has moved up-slope and grazing has increased at lower elevations in the past 100 years and distribution of bumblebee-visited plants shifted similar distances uphill. Climate change also provides areas suitable for agriculture higher up the mountain and both climate and land use change shift the climatic niche of flowering plants uphill [18].

Climate change is predicted to have a significant influence on the distribution of bumblebees across Europe, with most species expected to decline considerably in range [39]. High-elevation habitats are predicted to become increasingly important for maintaining the biodiversity of bumblebees, as they are likely to become refuges of colder temperatures that may no longer exist at lower elevations under different scenarios of climate change [39,40]. At a finer scale in the Swiss Alps, the bumblebee community was predicted to not only lose range and increase in elevation but also to become more homogenized under climate change [41]. We do not observe any clear loss of once abundant species from the bumblebee community but if elevation increases continue, the more generalist (in habitat use and flower visitation)

bumblebees will likely begin to occupy the same space as the more specialist species, which often results in the decline in abundance of specialist species [42]. Furthermore, potential land use change alongside climate change is likely to make these refuges even smaller and more important [43], although this remains to be explicitly tested. The limited information available from the nineteenth century suggests significant land use changes have occurred in the region. In the nearby commune of Villelongue, also part of the Pyrenees National Park, pastures have increased, at elevations around and below 1000 m, from 4.9% to 25.8% of total surface area between 1950 and 2003 of which the majority was the conversion of meadows [22]. The loss of meadows and woodland at lower elevations to pasture removes necessary resources and may result in the decrease in populations of wild pollinators and cause species to follow suitable habitat to higher elevations. Agricultural intensification is one of the leading causes of decline in wild pollinator populations [44]. In addition, in 2005 and 2006 significant grazing was observed at lower elevations, which may explain why fewer species were observed at 1000 m than in 1889.

Climate and land use change can potentially explain the shifts observed in bumblebees, both by direct pressure and indirectly by driving the loss or movement of floral resources [45]. Bumblebee species provide pollination services for many wildflower species. Due to their preference for cold conditions and large foraging range, bumblebees are often vital pollinators for plants that exist in cold, unpredictable climates and in fragmented habitats [12]. A good example of this are the pollination services bumblebees provide in European mountain habitats including the Pyrenees [46]. Different rates of elevation range change and phenological shifts caused by climate change between plants and their pollinators may decrease the effectiveness of this service and in the case of specialists may result in significant population declines [19]. The available data allow us to reach a preliminary conclusion regarding differences in elevation range between bumblebees and the plants they visit. The results suggest that, as most bumblebee species are generalist visitors, they have the potential to follow suitable resources to higher elevations but this is unlikely to be the only driver, except maybe for bumblebees specialized in their feeding. Focussing on the three most visited plants we see that single plant species (or in the case of *Aconitum*, genera) can potentially support a large community of bumblebees. For example, species of *Aconitum* can be adapted to be pollinated by a variety of bumblebee species, especially those with long tongues [47]. Therefore, as a genus which is occurring at higher elevations within the region it may become a more common food source for a variety of bumblebees as they shift to higher elevations due to environmental changes. The same is true of the *C. defloratus* and *C. eriophorum* species, and many Asteraceae [12], which were also visited by a diverse group of bumblebees at higher elevations. In general, we see a greater shift in the bumblebee-visited plants than in the bumblebees themselves. This suggests that bumblebees may be able to adapt to new feeding resources at the same elevation while also tracking former floral resources higher uphill, resulting in a lag. This lag may be because bumblebees have specific nesting requirements and the establishment of new colonies at higher elevations may take time. A repeated survey within the next decade would allow this hypothesis to

be tested. This result is contrary to the results observed in a similar study in the Rocky Mountains where they found that the plants most visited by bumblebees did not shift in elevations over 33-years [35]. An additional driver of pollinator behaviour and potential range shifts is flowering phenology, which we did not directly test in this study. Decreased temporal overlap is expected at high elevations, when plant species flower earlier in the year and the corresponding behavioural shift by bumblebees is either lagging or absent [35]. This could mediate range shifts by either forcing bumblebees to change their food preferences or adapt to visiting the same plants at higher elevations where temperatures are colder.

*Bombus gerstaeckeri*, a red-listed vulnerable species, is the only feeding specialist (on *Aconitum*) of the observed assemblage and showed a large increase in elevation [48,49]. In the surveys, *Aconitum* showed an increase in elevation of approximately 400 m. This suggests that *B. gerstaeckeri* may have been driven to higher elevations to maintain access to its exclusive food source because of climate and land use changes. We see similar patterns for *B. wurflenii*, which showed a clear preference for *Aconitum* in 2005–06. A narrow diet is likely to increase the vulnerability of bumblebees to drivers of decline [38,50]. Species specialized in high-elevation areas are likely to suffer more from climate change than other species [51]. Additionally, *Aconitum* had a far greater number of visitors in 2005–06 at higher elevation, suggesting that competition may have increased.

Here, we show a comparison between two distant periods. However, the most optimal way to measure insect decline and distributional shifts in the Pyrenees and elsewhere is to produce long-term time-series data with continued monitoring of communities; this can produce robust trends directly linked to proposed drivers of decline [52]. We show that there has been a distributional shift of bumblebees and the plants they visit, however, to better understand community dynamics, long-term time-series data would allow researchers to examine yearly variations, shifts in phenology and changes to population densities at different elevations. The results presented here show that this is urgently needed to understand how best to conserve important, high elevation bumblebee communities.

Data accessibility. The datasets and scripts supporting this article are available for download at https://github.com/lmar116/Pyrenees Bumblebees.

Authors' contributions. J.C.B. and W.K. conceived of the idea and planned the experimental design. J.C.B. carried out the field sampling. J.C.B., F.P. and L.M. extracted and prepared the historical database. S.R. and J.C.B. identified the specimens. L.M., F.P., J.C.B. and N.D. planned the data analysis. L.M. and F.P. prepared the data and carried out the data analysis. L.M. took the lead in writing the manuscript. All authors discussed the results and contributed to the final manuscript.

Competing interests. The authors declare no competing interests.

Funding. This work was funded by the EU through projects ALARM (Assessing Large-scale environmental Risks for biodiversity with tested Methods, www.alarmproject.net, EU 6th Framework Program) and STEP (Status and Trends of European Pollinators, www.step-project.net, EU 7th Framework Program). This work was also supported in part by BELSPO funded BELBEES project (BR/132/A1/BELBEES; Multidiciplinary assessment of BELgian wild BEE decline to adapt mitigation management policy).

Acknowledgements. We thank Judith Slaa, Simon Bryant and Chris Biesmeijer for assistance during the fieldwork and Zjef Pereboom for library assistance.

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
