## [Reviewer comments · Proceedings of the Royal Society B: Biological Sciences]

Review History

RSPB-2020-0791.R0 (Original submission)

Review form: Reviewer 1

Recommendation

Major revision is needed (please make suggestions in comments)

Scientific importance: Is the manuscript an original and important contribution to its field?

Good

General interest: Is the paper of sufficient general interest?

Good

Quality of the paper: Is the overall quality of the paper suitable?

Good

Is the length of the paper justified?

Yes

Should the paper be seen by a specialist statistical reviewer?

No

Do you have any concerns about statistical analyses in this paper? If so, please specify them explicitly in your report.

No

It is a condition of publication that authors make their supporting data, code and materials available - either as supplementary material or hosted in an external repository. Please rate, if applicable, the supporting data on the following criteria.

Is it accessible?

No

Is it clear?

N/A

Is it adequate?

N/A

Do you have any ethical concerns with this paper?

No

Comments to the Author

The manuscript by Marshall et al presents an interesting study that aimed at describing changes in elevation of bumblebees and the plants they visit in a mountainous area, by replicating a century-old sampling. I appreciate a lot this kind of attempts to re-analyse / replicate old surveys since it is probably the only way to get a picture of biodiversity changes at such a large temporal scale (besides paleoecological data). However, the challenge is always to make sure that data collected with different research traditions, various protocols and uncertain locations remain comparable. Here, we are still lacking some details of both the present and historical surveys to be certain that this is the case.

First, how and where the historical collection was carried out is not entirely clear. The text mentions seven locations, but their names do not correspond to those shown on the map Fig 1. Moreover, Fig1 also shows 10 buffer areas. There are here discrepancies that should be corrected or explained. We also don't know the sampling effort of the historical survey. I understand that it may be impossible to give accurate numbers of time spent in the field, but we should at least have a broad idea if data come from a single short visit or from several months of extensive field work aimed at collecting every possible species.

Second, I also struggled to understand exactly how you chose the 2005/06 sites: many of them seem to be quite far from the areas sampled in 1889 (not even in the buffers) and, from my anecdotal knowledge of the area, you also seem to have ignored the historical areas at the highest altitudes (no site in the cirque de Gavarnie / brèche de Roland area). I am sure there are logical and consistent rules for site selection that can be explained.

Beside those points, I also got the feeling that your analysis of changes in bumblebee and plant elevation was somewhat disconnected from all your work on LULC and temperature change. You seem to have done a hard work to describe changes in LULC and climate, but at this point these results are only little connected to your findings (only a few sentences in the discussion). Wasn't there a better way to test how changes in bumblebee or plant communities were linked to these drivers?

Other, minor, comments:

L31, and perhaps L 90: Being a rather small mountain range, I think it may not be obvious to non-European people where the Pyrenees are. Perhaps you should just add, at least in the abstract,

that it is located in southwest Europe, or at the French/Spanish border.

L 74 : What does “are adapted to deal with lower temperatures” mean precisely? Do you mean many bumblebee species have preadaptation to temperatures lower than what they currently experience? Or do you mean they have a wide climatic niche? Or maybe this is just a typo and you meant “adapted to low temperatures”.

L126-138: It is very unclear how you quantified LULC change. It seems from Fig S3 that you extracted present and past LULC in buffer areas around sampling locations, but nothing like that is said in the text.

L 231-233: What is the % after species names? Is it the percentage of this species relative to total bumblebee abundance?

L 337: You write that you did not observe homogenisation of bumblebee communities, but you did not test it (or at least you did not present an analysis that supports this statement).

Fig1: It would be easier to visualize also here the altitude of sampling plots, for example by having varying colours for collection locations depending on their altitude.

Fig2: Please be more specific about how curves have been extrapolated. It seems the rule was to extrapolate to 2 times the number of sampled individuals. If it is the case, please write it.

Fig3: Why does the caption say that stars refer to significance in linear models, while those linear models are not described in text and seem to have been computed only for illustration purposes in that figure? You already gave significance of Pearson correlations, no need to add confusion with linear models.

Review form: Reviewer 2

Recommendation

Major revision is needed (please make suggestions in comments)

Scientific importance: Is the manuscript an original and important contribution to its field?

Acceptable

General interest: Is the paper of sufficient general interest?

Acceptable

Quality of the paper: Is the overall quality of the paper suitable?

Marginal

Is the length of the paper justified?

Yes

Should the paper be seen by a specialist statistical reviewer?

No

Do you have any concerns about statistical analyses in this paper? If so, please specify them explicitly in your report.

No

It is a condition of publication that authors make their supporting data, code and materials available - either as supplementary material or hosted in an external repository. Please rate, if applicable, the supporting data on the following criteria.

Is it accessible?

N/A

Is it clear?

N/A

Is it adequate?

N/A

Do you have any ethical concerns with this paper?

No

Comments to the Author

The authors demonstrate range shifts of bumble bees and the plants they visit between two surveys 115 years apart. This work may add additional information to the field of species redistribution and bumble bee ecology. My concern about the manuscript was that the results regarding plant-pollinator association have not been well discussed. The analysis of habitat changes has not yet directly support their arguments. The links between results and discussion can be largely improved.

“The region has warmed significantly over this period, alongside shifts in agricultural land use and forest.” – Can you specify information in the abstract? How would land use change affect the range shifting?

Most of the citations in the introduction need to be updated. “...species from these cooler areas ‘falling off the top of the mountain’ as suitable conditions no longer exist and as species, which dominate in warmer areas, outcompete them[6].” – Freeman et al. 2018 PNAS would be a more relevant paper to cite. “Climate change impacts the spatial distribution of biodiversity[2], driving species to higher elevations and latitudes [3-5]” – Several recent review paper have comprehensively addressed this issue, such as Pecl et al. 2017 Science, Lenoir and Svenning 2015 Ecography. “...the differences in the direction or speed of range shifts could lead to disruptions of interaction networks[4].” – This work does not really focus on species interaction.

The insect observation appears to be a by-product of plan survey in the original study. It is difficult to tell whether the protocol of two surveys are comparable, including the collection of bumble bee abundance and flowering density. Please provide more details of the field work and how the authors define each visit and calculate the visitation frequency.

It doesn't make much sense to compare proportional distribution between years and at the three elevational ranges. The authors may exclude Fig.2 or move it to supplementary.

Fig.3 c-d shows the relationship between the elevation shift in bumblebees and the plants they visited. It combines diverse information and is difficult to interpret. The bumble bees visited different plant species of which all shift at different rates. The authors may consider the strengths of pollinating association with different plant species and then disentangle the range shifts with different degree of association.

I would expect more discussion on the changes of visitation profiles of frequently visited plants. The visiting pollinators change dramatically with time and elevational span. This may provide rich information about the diverse range shifts.

The discussion about land use change was lengthy but not based on any specific result demonstrating how habitat change may have contributed to the observed range shifts. Information of habitat change reveals not much of its impacts, particularly meadows/pastures cannot be distinguished from grassland.

I would not recommend using “spatial mismatch” to represent the different rates of range shifts.

Lastly, flowering phenology is crucial for pollinators. Please address how it would mediate the visiting behavior and interact with observed range shifts.

Decision letter (RSPB-2020-0791.R0)

14-May-2020

Dear Dr Marshall,

I am writing to inform you that your manuscript RSPB-2020-0791 entitled "Bumblebees moving up: shifts in elevation ranges in the Pyrenees over 115 years" has, in its current form, been rejected for publication in Proceedings B.

This action has been taken on the advice of referees, who have recommended that substantial revisions are necessary. With this in mind we would be happy to consider a resubmission, provided the comments of the referees are fully addressed. However please note that this is not a provisional acceptance.

The resubmission will be treated as a new manuscript. However, we will approach the same reviewers if they are available and it is deemed appropriate to do so by the Editor. Please note that resubmissions must be submitted within six months of the date of this email. In exceptional circumstances, extensions may be possible if agreed with the Editorial Office. Please make sure to address the requirements for data archiving too.

Finally, I hope you and your co-authors are well in these challenging times.

Sincerely,
Professor Loeske Kruuk
mailto: proceedingsb@royalsociety.org

Associate Editor

Comments to Author:

I have now received two reviews from two reviewers who are extremely well-qualified to assess the manuscript. Both see promise in the work but both raised significant and consistent concerns, which prevent the paper from moving forward at PRSB in its current form. If the authors feel they can address all concerns raised I would welcome a new submission, which would go out to review again. The most important issues are 1) better explain the methods and site selection logic between the two surveys to demonstrate that they are in fact directly comparable. Direct comparability between the two time points is the crux of any resurvey paper. 2) Both reviewers raised concerns that the various components of the study (abiotic change, bees, plants) were not sufficiently connected. In my opinion it is the combination of plants and bees that makes this study really novel and interesting, so tying these aspects together (more info about sampling regimes, what is really being captured in the data, justification of conclusions) is key.

Thank you for submitting your work to PRSB - I hope you are all doing well during the pandemic.

Reviewer(s)' Comments to Author:

Referee: 1

Comments to the Author(s)

The manuscript by Marshall et al presents an interesting study that aimed at describing changes in elevation of bumblebees and the plants they visit in a mountainous area, by replicating a century-old sampling. I appreciate a lot this kind of attempts to re-analyse / replicate old surveys since it is probably the only way to get a picture of biodiversity changes at such a large temporal scale (besides paleoecological data). However, the challenge is always to make sure that data collected with different research traditions, various protocols and uncertain locations remain comparable. Here, we are still lacking some details of both the present and historical surveys to be certain that this is the case.

First, how and where the historical collection was carried out is not entirely clear. The text mentions seven locations, but their names do not correspond to those shown on the map Fig 1. Moreover, Fig1 also shows 10 buffer areas. There are here discrepancies that should be corrected or explained. We also don't know the sampling effort of the historical survey. I understand that it may be impossible to give accurate numbers of time spent in the field, but we should at least have a broad idea if data come from a single short visit or from several months of extensive field work aimed at collecting every possible species.

Second, I also struggled to understand exactly how you chose the 2005/06 sites: many of them seem to be quite far from the areas sampled in 1889 (not even in the buffers) and, from my anecdotal knowledge of the area, you also seem to have ignored the historical areas at the highest altitudes (no site in the cirque de Gavarnie / brèche de Roland area). I am sure there are logical and consistent rules for site selection that can be explained.

Beside those points, I also got the feeling that your analysis of changes in bumblebee and plant elevation was somewhat disconnected from all your work on LULC and temperature change. You seem to have done a hard work to describe changes in LULC and climate, but at this point these results are only little connected to your findings (only a few sentences in the discussion). Wasn't there a better way to test how changes in bumblebee or plant communities were linked to these drivers?

Other, minor, comments:

L31, and perhaps L 90: Being a rather small mountain range, I think it may not be obvious to non-European people where the Pyrenees are. Perhaps you should just add, at least in the abstract, that it is located in southwest Europe, or at the French/Spanish border.

L 74 : What does “are adapted to deal with lower temperatures” mean precisely? Do you mean many bumblebee species have preadaptation to temperatures lower than what they currently experience? Or do you mean they have a wide climatic niche? Or maybe this is just a typo and you meant “adapted to low temperatures”.

L126-138: It is very unclear how you quantified LULC change. It seems from Fig S3 that you extracted present and past LULC in buffer areas around sampling locations, but nothing like that is said in the text.

L 231-233: What is the % after species names? Is it the percentage of this species relative to total bumblebee abundance?

L 337: You write that you did not observe homogenisation of bumblebee communities, but you did not test it (or at least you did not present an analysis that supports this statement).

Fig1: It would be easier to visualize also here the altitude of sampling plots, for example by having varying colours for collection locations depending on their altitude.

Fig2: Please be more specific about how curves have been extrapolated. It seems the rule was to extrapolate to 2 times the number of sampled individuals. If it is the case, please write it.

Fig3: Why does the caption say that stars refer to significance in linear models, while those linear models are not described in text and seem to have been computed only for illustration purposes in that figure? You already gave significance of Pearson correlations, no need to add confusion with linear models.

Referee: 2

Comments to the Author(s)

The authors demonstrate range shifts of bumble bees and the plants they visit between two surveys 115 years apart. This work may add additional information to the field of species redistribution and bumble bee ecology. My concern about the manuscript was that the results regarding plant-pollinator association have not been well discussed. The analysis of habitat changes has not yet directly support their arguments. The links between results and discussion can be largely improved.

“The region has warmed significantly over this period, alongside shifts in agricultural land use and forest.” – Can you specify information in the abstract? How would land use change affect the range shifting?

Most of the citations in the introduction need to be updated. “...species from these cooler areas ‘falling off the top of the mountain’ as suitable conditions no longer exist and as species, which dominate in warmer areas, outcompete them[6].” – Freeman et al. 2018 PNAS would be a more relevant paper to cite. “Climate change impacts the spatial distribution of biodiversity[2], driving species to higher elevations and latitudes [3-5]” – Several recent review paper have comprehensively addressed this issue, such as Pecl et al. 2017 Science, Lenoir and Svenning 2015 Ecography. “...the differences in the direction or speed of range shifts could lead to disruptions of interaction networks[4].” – This work does not really focus on species interaction.

The insect observation appears to be a by-product of plan survey in the original study. It is difficult to tell whether the protocol of two surveys are comparable, including the collection of bumble bee abundance and flowering density. Please provide more details of the field work and how the authors define each visit and calculate the visitation frequency.

It doesn't make much sense to compare proportional distribution between years and at the three elevational ranges. The authors may exclude Fig.2 or move it to supplementary.

Fig.3 c-d shows the relationship between the elevation shift in bumblebees and the plants they visited. It combines diverse information and is difficult to interpret. The bumble bees visited different plant species of which all shift at different rates. The authors may consider the strengths of pollinating association with different plant species and then disentangle the range shifts with different degree of association.

I would expect more discussion on the changes of visitation profiles of frequently visited plants. The visiting pollinators change dramatically with time and elevational span. This may provide rich information about the diverse range shifts.

The discussion about land use change was lengthy but not based on any specific result demonstrating how habitat change may have contributed to the observed range shifts. Information of habitat change reveals not much of its impacts, particularly meadows/pastures cannot be distinguished from grassland.

I would not recommend using "spatial mismatch" to represent the different rates of range shifts.

Lastly, flowering phenology is crucial for pollinators. Please address how it would mediate the visiting behavior and interact with observed range shifts.

Author's Response to Decision Letter for (RSPB-2020-2201.R0)

See Appendix A.

RSPB-2020-2201.R1 (Revision)

Review form: Reviewer 1

Recommendation

Accept with minor revision (please list in comments)

Scientific importance: Is the manuscript an original and important contribution to its field?

Good

General interest: Is the paper of sufficient general interest?

Good

Quality of the paper: Is the overall quality of the paper suitable?

Good

Is the length of the paper justified?

Yes

Should the paper be seen by a specialist statistical reviewer?

No

Do you have any concerns about statistical analyses in this paper? If so, please specify them explicitly in your report.

No

It is a condition of publication that authors make their supporting data, code and materials available - either as supplementary material or hosted in an external repository. Please rate, if applicable, the supporting data on the following criteria.

Is it accessible?

No

Is it clear?

N/A

Is it adequate?

N/A

Do you have any ethical concerns with this paper?

No

Comments to the Author

I have reviewed the revision of the manuscript "Bumblebees moving up: shifts in elevation ranges in the Pyrenees over 115 years", and observed that the authors satisfactorily addressed most of my comments. I appreciate the new analysis of the shifts in climate ranges, and the fact that the potential effect of land-use/land-cover change is downplayed in the discussion, being largely hypothetical at this point.

Thanks also for the additional explanations about site selection and historical collections. Your replication strategy sounds more explicit to me. Still, I think you could be even more precise, in the sense that you should make clear in the text, in the appropriate "1889 Collections" section, what level of information you had regarding historical sampling locations. I understand broadly that this was only approximate, so that your replication was mainly about finding a place at similar altitude, in the vicinity of the supposed location, with the same plants if possible. However, I am still struggling a bit to understand if you knew, e.g. the name of a road along which plants/pollinators were sampled, or an indication like "after walking 1h from this place", etc.

[Edit: after writing my comment, I searched online for MacLeod's publication and, although I don't speak Dutch, I was able to find how this information was recorded. It seems that for a given record you only have the name of the area, and an approximate altitude, e.g. P306 in MacLeod 1890: "Bombus lapidarius, Gèdre, 1200". It was not entirely clear to me that the name of the 7 areas was the only location information you had! I understand now even better the reasoning behind your sampling strategy. If it wasn't clear to me, maybe it could also be confusing for other readers.]

Other minor comments:

Fig 2: it may sound like a detail but I am definitely not a fan of p-values summarised by a star. Since you calculated it with precision, please write the p-value in full, otherwise it is unnecessarily hiding information.

L292: missing space between 15 and were

L330-331: another study by the same group (Fourcade et al 2019, Biodiversity & Conservation) also discussed altitudinal shift in bumblebees in Scandinavia. You don't necessarily have to cite it but I mention it anyway, in case you had missed it.

In conclusion, I believe this manuscript provides interesting results, relevant to both entomologists and climate change researchers. Anecdotally, I hiked this summer in the very same areas as described in this study and I couldn't help but think about it every time I encountered a bumblebee.

Review form: Reviewer 2

Recommendation

Accept as is

Scientific importance: Is the manuscript an original and important contribution to its field?

Good

General interest: Is the paper of sufficient general interest?

Good

Quality of the paper: Is the overall quality of the paper suitable?

Good

Is the length of the paper justified?

Yes

Should the paper be seen by a specialist statistical reviewer?

No

Do you have any concerns about statistical analyses in this paper? If so, please specify them explicitly in your report.

No

It is a condition of publication that authors make their supporting data, code and materials available - either as supplementary material or hosted in an external repository. Please rate, if applicable, the supporting data on the following criteria.

Is it accessible?

No

Is it clear?

N/A

Is it adequate?

N/A

Do you have any ethical concerns with this paper?

No

Comments to the Author

I think the author has responded to the reviewer's questions, made necessary corrections, and considerably enhanced the interpretation of this study. Adequate background information on the historical investigation and resurvey methodology has been provided so that the readers can evaluate the information and limitations of the findings. There is also better coherence in the analysis of the findings and their discussion.

Decision letter (RSPB-2020-2201.R0)

12-Oct-2020

Dear Dr Marshall

I am pleased to inform you that your manuscript RSPB-2020-2201 entitled "Bumblebees moving up: shifts in elevation ranges in the Pyrenees over 115 years" has been accepted for publication in Proceedings B.

The Associate Editor and referee have recommended publication, but have also suggested some minor revisions to your manuscript. Therefore, I invite you to respond to their comments and revise your manuscript. Because the schedule for publication is very tight, it is a condition of publication that you submit the revised version of your manuscript within 7 days. If you do not think you will be able to meet this date please let us know.

Sincerely,
Professor Loeske Kruuk
mailto: proceedingsb@royalsociety.org

Associate Editor
Board Member
Comments to Author:

Both reviewers were very positive about the resubmitted manuscript, so I am pleased to recommend a decision of accept with minor revision (ie it will not have to go out to review again). One reviewer would still like to see additional clarification in some places. Having read the manuscript carefully I would also suggest that the authors go through it carefully to make sure the wording is clear and precise (accurate) throughout. I've attached a list of examples below but this is not exhaustive.

Abstract

L27 - 're-wired' metaphor unclear

L35 - 'in accordance with' or more simply 'according to'. But since the causal relationship has not been established 'with' might be best

L36 - check grammar here

L38-39 - drivers of? impacts of?

Intro

L59 – ‘often’ driving species to higher... (there are many counter examples)

L62 – change ‘and species...’ to ‘or species... outcompete them’. Relative importance of the two not always clear

L67 – don’t think ‘conceal’ is the right word here. ‘Make up for’?

L73 – since you are discussing ecological responses more than evolutionary ones, change ‘adapt to’ to ‘respond to’

L79 – specify: ‘species distributions’ & ‘insect distributions’ since that’s what the examples are about (and the relevant comparison)

Fig 1

Change ‘expected’ to ‘estimated’

Reconcile singular/plural. ‘area’ & ‘areas’

Delete ‘grey’ (triangles are now green)

Climate Change and Land Use Changes

L141 – clarify what unit of observation was here. 1 data point per bee species (if so change wording at end to reflect this)? Or separate test for each species, in which case what was an observation, and how did you deal with spatial autocorrelation of climate pixels? (1st option seems more appropriate)

L143 – acronyms make for hard reading. Recommend replacing ‘LULC’ with simply ‘land use’ & explain that this shorthand for Land use/Land cover

Editor Comments to Author: Abstract, line 35/36: ‘provide preliminary evidence that some bumblebee species shift their ranges’ (i.e. avoid the risk of ‘some bumblebees shift’ sounding like individuals, rather than the species). See also the AE’s comments on the wording of this sentence.

Reviewer(s)’ Comments to Author:

Referee: 1

Comments to the Author(s).

I have reviewed the revision of the manuscript “Bumblebees moving up: shifts in elevation ranges in the Pyrenees over 115 years”, and observed that the authors satisfactorily addressed most of my comments. I appreciate the new analysis of the shifts in climate ranges, and the fact that the potential effect of land-use/land-cover change is downplayed in the discussion, being largely hypothetical at this point.

Thanks also for the additional explanations about site selection and historical collections. Your replication strategy sounds more explicit to me. Still, I think you could be even more precise, in the sense that you should make clear in the text, in the appropriate “1889 Collections” section, what level of information you had regarding historical sampling locations. I understand broadly that this was only approximate, so that your replication was mainly about finding a place at similar altitude, in the vicinity of the supposed location, with the same plants if possible.

However, I am still struggling a bit to understand if you knew, e.g. the name of a road along which plants/pollinators were sampled, or an indication like “after walking 1h from this place”, etc.

[Edit: after writing my comment, I searched online for MacLeod’s publication and, although I don’t speak Dutch, I was able to find how this information was recorded. It seems that for a given record you only have the name of the area, and an approximate altitude, e.g. P306 in MacLeod 1890: “*Bombus lapidarius*, Gèdre, 1200”. It was not entirely clear to me that the name of the 7 areas was the only location information you had! I understand now even better the reasoning behind your sampling strategy. If it wasn’t clear to me, maybe it could also be confusing for other readers.]

Other minor comments:

Fig 2: it may sound like a detail but I am definitely not a fan of p-values summarised by a star. Since you calculated it with precision, please write the p-value in full, otherwise it is unnecessarily hiding information.

L292: missing space between 15 and were

L330-331: another study by the same group (Fourcade et al 2019, Biodiversity & Conservation) also discussed altitudinal shift in bumblebees in Scandinavia. You don't necessarily have to cite it but I mention it anyway, in case you had missed it.

In conclusion, I believe this manuscript provides interesting results, relevant to both entomologists and climate change researchers. Anecdotally, I hiked this summer in the very same areas as described in this study and I couldn't help but think about it every time I encountered a bumblebee.

Referee: 2

Comments to the Author(s).

I think the author has responded to the reviewer's questions, made necessary corrections, and considerably enhanced the interpretation of this study. Adequate background information on the historical investigation and resurvey methodology has been provided so that the readers can evaluate the information and limitations of the findings. There is also better coherence in the analysis of the findings and their discussion.

Author's Response to Decision Letter for (RSPB-2020-2201.R0)

See Appendix B.

Decision letter (RSPB-2020-2201.R1)

20-Oct-2020

Dear Dr Marshall

I am pleased to inform you that your manuscript entitled "Bumblebees moving up: shifts in elevation ranges in the Pyrenees over 115 years" has been accepted for publication in Proceedings B.

Open Access

Paper charges

Sincerely,

Proceedings B

Appendix A

Dr. Leon Marshall

Agroecology Lab

Université Libre de Bruxelles

50 avenue F.D. Roosevelt, 1050 Bruxelles, Belgium

05-09-2020

Manuscript re-submission: “Bumblebees moving up: shifts in elevation ranges in the Pyrenees over 115 years” (RSPB-2020-0791) by Leon Marshall, Floor Perdijk, Nicolas Dendoncker, William Kunin, Stuart Roberts and Koos Biesmeijer.

Dear Editorial Team,

On behalf of my co-authors, I am pleased to re-submit to Proceedings B a revised version of our manuscript titled “Bumblebees moving up: shifts in elevation ranges in the Pyrenees over 115 years” (RSPB-2020-0791). We would like to express our gratitude to the editorial team and the referees for the constructive and relevant comments on the previous version of our manuscript and for the opportunity to submit a revised version.

We have now thoroughly revised the manuscript in response to the reviewers’ comments (see below for a point-by-point response to all comments). Specifically, we have updated the manuscript in response to the two key issues raised; (1) to improve the explanation of why the two sampling periods are directly comparable. In response to this we have provided a full description of the sites visited in 1889 as described by MacLeod, we made it clearer on the map where the majority of the sampling has occurred and we have increased the explanation of the thought processes and actions made in 2005-06 to best repeat the 1889 surveys. In addition (2) the need to improve the connection between the various components of the study (abiotic change, bees, plants) in the manuscript. In response to this, we have produced new figures and results examining the change in climatic ranges between the two periods and connected these to the shifts in resource use by bees. Furthermore, we now discuss the importance of observed shifts in plants and their bumblebee visitors in more detail.

We believe that the revision has significantly improved our manuscript and we hope it is now suitable for publication in Proceedings B.

Yours sincerely,

Leon Marshall (on behalf of all authors)

Reviewer(s)' Comments to Author:

Referee: 1

Comments to the Author(s)

Referee 1: The manuscript by Marshall et al presents an interesting study that aimed at describing changes in elevation of bumblebees and the plants they visit in a mountainous area, by replicating a century-old sampling. I appreciate a lot this kind of attempts to re-analyse / replicate old surveys since it is probably the only way to get a picture of biodiversity changes at such a large temporal scale (besides paleoecological data). However, the challenge is always to make sure that data collected with different research traditions, various protocols and uncertain locations remain comparable. Here, we are still lacking some details of both the present and historical surveys to be certain that this is the case.

RESPONSE: The main aim of our study was to compare the present state of bumblebees and plants in the Pyrenees with historic information. The historic information derived from the publication by Julius MacLeod from 1891: "De Pyreneënbloemen en hare bevruchting door insecten". MacLeod spent August 5 to 31 in 1889 in the area around Gèdre in the Hautes Pyrenees, France to assess which insects were visiting the flowering plants. Most time was spent between 1000m (around Gèdre) and 1500m (cascade de Gavarnie), with some visits to higher areas (Gavarnie up to 2300m, Cirque du Troumouse (up to 2000m) and one visit to the Brèche de Roland (2800m).

The short description of the area can be summarized as follows: The Luz valley around Gèdre consisted of pastures and arable fields, with some low shrubs on the hillsides, but no forest. The whole area from 900-1300 was very rich in flowers in August particularly on the gravel beds along the river and on the extensive rock beds ('chaos') in the area. The meadows were mostly mown off in August (normal practice being a first cut in June, second one in July and sometimes third cut in August).

Around the Gavarnie area they found flora with alpine character. At the base of the Cirque du Gavarnie (1600-1700m), they still found snow in August with abundant alpine flora. There were some pine and fir forests present between the alpine meadows. The presence of alpine flora at such a low elevation is a result of the high walls of Cirque du Gavarnie (400-500m high) on the south side, preventing warm southern winds and limiting sunlight reaching the soil.

The Cirque du Troumouse was visited from Gèdre following the road to Héas (around 1500m) and on to Troumouse (up to 2000m). The area was heavily grazed ('thousands of

sheep') and few flowers were left in Troumouse. Only poisonous Aconitums and spiny thistles (e.g. *Carduus carlinoides*) were flowering abundantly. Despite the overgrazing they observed abundant insect life of mostly small species.

The plateau of Saugué (1500-1650m) consisted of hay meadows ('the best in the region')

This extra information has been added as Appendix A in the supplementary material.

Referee 1: First, how and where the historical collection was carried out is not entirely clear. The text mentions seven locations, but their names do not correspond to those shown on the map Fig 1. Moreover, Fig1 also shows 10 buffer areas. There are here discrepancies that should be corrected or explained. We also don't know the sampling effort of the historical survey. I understand that it may be impossible to give accurate numbers of time spent in the field, but we should at least have a broad idea if data come from a single short visit or from several months of extensive field work aimed at collecting every possible species.

RESPONSE: The map has been edited to better represent the information available from MacLeod in 1889. In addition, we have added some text explaining how we have tried to maximize the match between the historic sites and our own recordings, lines 175-186.

"In 2005 (8th to 25th August) and 2006 (14th to 31st August), two surveys were conducted to analyse the plant visitor community of the most visited plant species in 1889, in the same areas that MacLeod sampled. The fundamental difference between the two surveys is the target organism. In 2005-06 the target was the plant visitors, and a selection of plants was made first based on MacLeod's findings, i.e. observing the same plant species that were observed by MacLeod. However, to maximize sampling of the pollinator community, additional observations were made on other well-visited abundant flowering plant species. Therefore, direct comparisons of whole networks are not possible. In summary, the first goal was to find a match with the location and plant species observed by MacLeod. If that was not possible, an alternative nearby location at similar elevation with the same plant species observed by MacLeod was searched for. In addition, visitors of other flowering plants at the original MacLeod locations were observed and lastly, other flowering plant species were observed at locations corresponding in elevation, but different from the MacLeod locations. In this way, we strived to make the 1889 and 2005-6 studies as comparable as possible in terms of visited plants, locations and elevations."

We trust that this clarifies the sampling issues mentioned by the reviewer.

Referee 1: Second, I also struggled to understand exactly how you chose the 2005/06 sites: many of them seem to be quite far from the areas sampled in 1889 (not even in the buffers) and, from my anecdotal knowledge of the area, you also seem to have ignored the historical areas at the highest altitudes (no site in the cirque de Gavarnie / brèche de Roland area). I am sure there are logical and consistent rules for site selection that can be explained.

RESPONSE: On the selection of the 2005-6 sites, we refer to the above reply.

On the high-altitude sites of MacLeod: While included as sites within his publication and therefore shown on the map, cirque de Gavarnie and the Breche de Roland constitute only 2 and 1 bumblebee records as this was not an area he spent considerable time surveying. Furthermore, in 2005-06 this area was inaccessible and could not be surveyed; this was not considered an issue at the time as this area did not represent the majority of MacLeod's surveys. In addition, we have sampled along the road to the nearby ski resort and beyond, which did not exist in MacLeod's time, but provides access to similar elevations as most of his high elevation sites.

We have edited the map to better reflect this discrepancy, we have fit large buffers around all areas he mentions in his book (see above and supplementary material). It is not clear in which directions he travelled, therefore, to illustrate this uncertainty we have taken 3km buffers around these points and presented them in grey. There are also areas which are far more common in the book and due to the elevation provided and the locations of the landmarks he mentions we can be more accurate with our mapping of these areas. These areas are presented as dark 2km buffer zones. These areas constitute 87% of the records he collected. See figure 1.

Referee 1: Beside those points, I also got the feeling that your analysis of changes in bumblebee and plant elevation was somewhat disconnected from all your work on LULC and temperature change. You seem to have done a hard work to describe changes in LULC and climate, but at this point these results are only little connected to your findings (only a few sentences in the discussion). Wasn't there a better way to test how changes in bumblebee or plant communities were linked to these drivers?

RESPONSE: Indeed, this is a difficulty that we encountered when analysing this dataset. Firstly, regarding land use, two factors make it impossible to be more detailed than comparing time periods at the regional scale. Firstly, MacLeod was not sufficiently clear in his records to extract exact land use information for each species record and secondly, the available land use information from the early part of the 20th century is not at a fine enough thematic or spatial resolution to attribute information to specimens.

Regarding climate there was indeed room to improve the connection between specimen records and climate observations. In the revised manuscript we did this by comparing the climate range for all observations of each species in both time periods and then statistical comparing whether the climate range occupied in 2005-06 is warmer than 1889. Please see figure 2 and lines 236-242. The results suggest that like with the plants the climate range has shifted further uphill than the bumblebees suggesting a potential lag. However, we observed five species that seemed to have maintained the same climate range in both periods.

“The majority of Bumblebee species showed a lower overall increase of their occupied elevation ranges in the average August temperature than the overall climate change modeled for the region (2.1°C). The total average mean difference in August temperatures of the occupied ranges of all species was 1.3°C. Bombus soroensis showed the greatest difference, occupying a climate range

3.4°C warmer on average. The smallest change was observed for B. gerstaeckeri, 0.24°C. Five species show no statistical difference ($p>0.05$) between their climate ranges in both periods. Including, B. gerstaeckeri, B. wurflenii, B. ruderarius, B. mesomelas and B. lapidarius (Figure 2)."

Referee 1: Other, minor, comments:

Referee 1: L31, and perhaps L 90: Being a rather small mountain range, I think it may not be obvious to non-European people where the Pyrenees are. Perhaps you should just add, at least in the abstract, that it is located in southwest Europe, or at the French/Spanish border.

RESPONSE: We have added this extra information into the abstract. Lines 31-32.

Referee 1: L 74 : What does "are adapted to deal with lower temperatures" mean precisely? Do you mean many bumblebee species have preadaptation to temperatures lower than what they currently experience? Or do you mean they have a wide climatic niche? Or maybe this is just a typo and you meant "adapted to low temperatures".

RESPONSE: Indeed, we mean adapted to low temperatures. Thank you for noticing this error. This has been changed in the manuscript. Line 76.

Referee 1: L126-138: It is very unclear how you quantified LULC change. It seems from Fig S3 that you extracted present and past LULC in buffer areas around sampling locations, but nothing like that is said in the text.

RESPONSE: We have clarified this in the text. Lines 143-145: *"We extracted the land use within each elevation profile at a 1 × 1km grid resolution for a 10km buffer surrounding the centroid of all collection records within the same elevation profile."*

Referee 1: L 231-233: What is the % after species names? Is it the percentage of this species relative to total bumblebee abundance?

RESPONSE: Yes, this is the correct interpretation of the percentage. We have added this to the first parentheses to make this abundantly clear. Lines 257-258.

Referee 1: L 337: You write that you did not observe homogenisation of bumblebee communities, but you did not test it (or at least you did not present an analysis that supports this statement).

RESPONSE: Indeed, homogenization is not the correct term here as that would imply a more detailed analysis on the abundance of each species. We meant to say that there does not seem to be any obvious loss of species from the region that we could have reliably measured. We have edited the manuscript to reflect this. Lines 390-391. *“We do not observe any clear loss of once abundant species from the bumblebee community”*

Referee 1: Fig1: It would be easier to visualize also here the altitude of sampling plots, for example by having varying colours for collection locations depending on their altitude.

RESPONSE: We have edited figure 1 considerably. MacLeod recorded his areas based on nearby landmarks and elevation therefore we cannot be sure of exact points, therefore we leave these areas as buffer regions. In order to illustrate elevation, we have coloured the 2005-06 records according to elevation which shows the variation across the region.

Referee 1: Fig2: Please be more specific about how curves have been extrapolated. It seems the rule was to extrapolate to 2 times the number of sampled individuals. If it is the case, please write it.

RESPONSE: Due to comments by Reviewer2 this figure is now considered as supplementary material. However, we have added the extrapolation information into the figure caption. The reviewer is indeed correct we used the iNext package with default settings which is to extrapolate species richness ‘up to double the reference sample’, beyond this the bias is too large for the result to be meaningful.

Referee 1: Fig3: Why does the caption say that stars refer to significance in linear models, while those linear models are not described in text and seem to have been computed only for illustration purposes in that figure? You already gave significance of Pearson correlations, no need to add confusion with linear models.

RESPONSE: In the case of these simple linear models the significance of Pearson correlations and the fixed effect from the linear model are equivalent. We have chosen to just present the results from the linear model as this provides more information.

Referee: 2

Comments to the Author(s)

The authors demonstrate range shifts of bumble bees and the plants they visit between two surveys 115 years apart. This work may add additional information to the field of species redistribution and bumble bee ecology. My concern about the manuscript was that the results regarding plant-pollinator association have not been well discussed. The analysis of

habitat changes has not yet directly support their arguments. The links between results and discussion can be largely improved.

Referee 2: “The region has warmed significantly over this period, alongside shifts in agricultural land use and forest.”— Can you specify information in the abstract? How would land use change affect the range shifting?

RESPONSE: We hypothesize that this is the result of loss of habitat at lower elevations, mostly due to intensification of agriculture and loss of semi-natural meadows, as observed in other areas of the Pyrenees. However, we cannot test this explicitly, please see response below regarding land use. Because we have decreased the focus on land use change, we do not think it is appropriate to speculate in the abstract beyond the fact that the land use has shifted in the region and neighbouring regions.

Referee 2: Most of the citations in the introduction need to be updated. “...species from these cooler areas ‘falling off the top of the mountain’ as suitable conditions no longer exist and as species, which dominate in warmer areas, outcompete them[6].” —Freeman et al. 2018 PNAS would be a more relevant paper to cite. “Climate change impacts the spatial distribution of biodiversity[2], driving species to higher elevations and latitudes [3-5] “— Several recent review paper have comprehensively addressed this issue, such as Pecl et al. 2017 Science, Lenoir and Svenning 2015 Ecography. “...the differences in the direction or speed of range shifts could lead to disruptions of interaction networks[4].” —This work does not really focus on species interaction.

RESPONSE: We have updated the citations in the introduction based on the reviewer's suggestions. We have replaced the case studies with the review papers mentioned.

Whilst we touch upon the importance of interactions between plants and bumblebees, we are not able to analyse our current data using network theory. Therefore, we have also decreased the emphasis on species interactions in the introduction.

Referee 2: The insect observation appears to be a by-product of plan survey in the original study. It is difficult to tell whether the protocol of two surveys are comparable, including the collection of bumble bee abundance and flowering density. Please provide more details of the field work and how the authors define each visit and calculate the visitation frequency.

RESPONSE: That the observations are a by-product is not a correct assessment of the referee. MacLeod’s main purpose was to understand which insects were visiting the plants in the area and to what extent. Therefore, he observed, counted (1,2,3, many) and caught all insects, and had them identified by the leading taxonomists. He then recorded the full list of interactions he observed per plant.

Clearly, we do not know how long he was observing plant species, but it is remarkable that all visitors including small flies and beetles, but also all bees and butterflies were recorded.

In fact, these studies by botanists (starting with Muller 1881, followed by Knuth 1895, Burkill and Willis 1895) and many others constitute studies with less bias in flower visitors than any observation by an entomologist, given that they are normally focussed on specific groups of visiting insects.

The best way we could match this was to observe patches of plants and locations that maximally matched his work (see remarks above – Reviewer1).

Referee2: It doesn't make much sense to compare proportional distribution between years and at the three elevational ranges. The authors may exclude Fig.2 or move it to supplementary.

RESPONSE: Thank you for this suggestion, we agree that the comparison is not necessarily appropriate for the main manuscript and have therefore shifted part of the figure to supplementary material and removed the other proportional comparison. See figure S4.

Referee 2: Fig.3 c-d shows the relationship between the elevation shift in bumblebees and the plants they visited. It combines diverse information and is difficult to interpret. The bumble bees visited different plant species of which all shift at different rates. The authors may consider the strengths of pollinating association with different plant species and then disentangle the range shifts with different degree of association.

RESPONSE: We thank the reviewer for the suggestion. We wanted to improve the interpretability of these figures for the reader and to more accurately represent the diet of the bumblebee species. The majority of the species are polylectic and visit a wide variety of plant species. Therefore, to, a priori, apply a strength of pollination association which would be interpolated from other areas seemed inappropriate. In order to still apply a more rigorous analysis of pollinator association we took into account the recorded associations across the two periods. For each species we calculated the mean shift of the visited plant species weighted by the total visitations of that bumblebee to that species. This more appropriately compares the elevation shift to the shift in plant species that we assume is a more targeted diet choice. We present this as a single figure 3c. with a fitted model. This shows that while there is a suggestion of a relationship, it is not strong enough to say that plants are shifting in line with their preferred plants. We do however see that there are examples of where this relationship is strong for some species and non-existent for others. We discuss this in detail on lines 215-219 as well as lines 297-304:

“To test if the elevation shift in bumblebees was comparable to shifts in the plants they visit, we calculated the mean elevation shift of the plants visited for each species. To reflect floral preferences between species we calculated a weighted mean, weighted by the frequency of observations on each plant. We then tested using a linear model whether there was a positive correlation between the elevation shift of bumblebees and the elevation shift of the plants they visit in the region.”

“The shift in bumblebees and the plants they visited in either period was weakly positively correlated (Linear regression: $R^2=0.24$, $\beta=0.25$, $p\text{-value}=0.1$; Fig 3c, d). Three species show a negative

relationship, B. sichelii, B. hortorum and B. soroensis. These species decreased in overall elevation whilst the plant species they visited in 1889 shifted between 200-350m uphill. The positive relationship observed is also dependent on the relationships between B. wurflenii, B. gerstaeckeri and Aconitum spp. with all three showing shifts over 400m. Similar shifts in elevation of bumblebees and the plants they visited were observed for B. pascuorum, and B. mesomelas. Overall, plant species shifted further than all bumblebee species except B. lapidarius.”

Referee 2: I would expect more discussion on the changes of visitation profiles of frequently visited plants. The visiting pollinators change dramatically with time and elevational span. This may provide rich information about the diverse range shifts.

RESPONSE: We have added extra information regarding this aspect of the study to the discussion. Indeed, we believe that this relationship shows strongly both the generalist nature of the bumblebees but also of the plants visited, which suggests a dynamic community and interaction network. This means the associations between specific plants and bees is unlikely to be the sole driver of elevational shifts. See lines 413-422.

“The results suggest that, as most bumblebee species are generalist visitors, they have the potential to follow suitable resources to higher elevations but this is unlikely to be the only driver except maybe for bumblebees specialised in their feeding. Focussing on the three most visited plants we see that single plant species (or in the case of Aconitum, genus) can potentially support a large community of bumblebees. For example, species of Aconitum can be adapted to be pollinated by a variety of bumblebee species, especially those with long tongues [48]. Therefore, as a genus which is occurring at higher elevations within the region it may become a more common food source for a variety of bumblebees as they shift to higher elevations due to environmental changes. The same is true of the two thistle species, and many Asteraceae in general [12], which were also visited by a diverse group of bumblebees at higher elevations.”

Referee 2: The discussion about land use change was lengthy but not based on any specific result demonstrating how habitat change may have contributed to the observed range shifts. Information of habitat change reveals not much of its impacts, particularly meadows/pastures cannot be distinguished from grassland.

RESPONSE: This is true, while we believe that land use is likely to have played a significant role in the shift of species from lower elevations this conclusion is based on anecdotal evidence from the site and is linked to land use change observed in other areas along the mountain range. The analysis conducted is not able to establish clear patterns or mechanisms due to thematic and spatial resolution of land use data available in the 19th century and early 20th century. For this reason, we have taken the reviewers advice and decreased the focus on land use in the discussion and only represent it as a possible hypothesis yet to be tested. See lines 393-396.

Referee 2: I would not recommend using “spatial mismatch” to represent the different rates of range shifts.

RESPONSE: We have changed this. See lines 410-413.

“Different rates of elevation range change and phenological shifts caused by climate change between plants and their pollinators may decrease the effectiveness of this service and in the case of specialists will result in significant population declines [19]. The available data allows us to reach a preliminary conclusion regarding differences in elevation range.”

Referee 2: Lastly, flowering phenology is crucial for pollinators. Please address how it would mediate the visiting behavior and interact with observed range shifts.

RESPONSE: We have expanded our discussion on this aspect. See lines 429-434.

“An additional driver of pollinator behaviour and potential range shifts is flowering phenology, which we did not directly test in this study. Decreased temporal overlap is expected at high elevations. Plant species flower earlier in the year and the corresponding behavioural shift by bumblebees is either lagging or absent[34]. This could mediate range shifts by either forcing bumblebees to change their food preferences or adapt to visiting the same plants at higher elevations where temperatures are colder.”

Extra information:

During the review process, detailed re-reading of MacLeod’s provided us with extra observations of bumblebee species where he clearly describes species as being found throughout an elevation range, where previously our data-mining had recorded these as present at only the lowest and highest elevations of the range mentioned. The inclusion of these observations does not change the interpretation of the results but does slightly decrease the overall mean elevation shifts of bumblebees and the plants they visited. This means the results in the present version of the manuscript are slightly more conservative than those presented before.

Appendix B

UNIVERSITÉ LIBRE DE BRUXELLES,
UNIVERSITÉ D'EUROPE

Dr. Leon Marshall

Agroecology Lab

Université Libre de Bruxelles

50 avenue F.D. Roosevelt, 1050 Bruxelles, Belgium

19-10-2020

Manuscript re-submission: “Bumblebees moving up: shifts in elevation ranges in the Pyrenees over 115 years” (RSPB-2020-0791) by Leon Marshall, Floor Perdijk, Nicolas Dendoncker, William Kunin, Stuart Roberts and Koos Biesmeijer.

Dear Editorial Team,

We would like to thank you for accepting our manuscript (RSPB-2020-0791) entitled “**Bumblebees moving up: shifts in elevation ranges in the Pyrenees over 115 years**” by Marshall et al. We have responded to and included all the final comments of the editors and reviewers. Please see below for a point by point response. In addition to this we have thoroughly proofread the manuscript to improve readability.

Yours sincerely,

Leon Marshall (on behalf of all authors)

Associate Editor

Board Member

Comments to Author:

AE: Both reviewers were very positive about the resubmitted manuscript, so I am pleased to recommend a decision of accept with minor revision (ie it will not have to go out to review again). One reviewer would still like to see additional clarification in some places. Having read the manuscript carefully I would also suggest that the authors go through it carefully to make sure the wording is clear and precise (accurate) throughout. I've attached a list of examples below but this is not exhaustive.

Response: Thank you for accepting our article, we agree that it has significantly improved thanks to the editor and reviewer's comments. We have responded to all comments, see below, fixing small errors in the manuscript and providing clarification where needed. Furthermore, we have thoroughly edited the grammar of the manuscript to ensure that it is clear and precise (accurate) throughout.

Abstract

AE: L27 – 're-wired' metaphor unclear

Response: We mean that the interacting species may change who they interact with. This message is just as clear by saying disrupted, so we have simply deleted 'or re-wired'.

AE: L35 – 'in accordance with' or more simply 'according to'. But since the causal relationship has not been established 'with' might be best

Response: We have changed it to 'with'

AE: L36 – check grammar here

Response: The sentence now reads: "We also observe that some species have been able to occupy the same climate range in both periods by shifting elevation range." See lines 35-37.

AE: L38-39 – drivers of? impacts of?

Response: This sentence has been edited. See lines 37-39:

"The results suggest the need for long-term monitoring to determine the role and impact of the different drivers of global change, especially in montane habitats where the impacts of climate changes are anticipated to be more extreme. "

Intro

AE: L59 – 'often' driving species to higher... (there are many counter examples)

Response: Changed as per suggestion.

AE: L62 – change 'and species...' to 'or species... outcompete them'. Relative importance of the two not always clear

Response: Changed as per suggestion.

AE: L67 – don't think 'conceal' is the right word here. 'Make up for'?

Response: Changed to 'counteract'.

AE: L73 – since you are discussing ecological responses more than evolutionary ones, change ‘adapt to’ to ‘respond to’

Response: Changed as per suggestion

AE: L79 – specify: ‘species distributions’ -> ‘insect distributions’ since that’s what the examples are about (and the relevant comparison)

Response: Changed as per suggestion.

AE: Fig 1

Change ‘expected’ to ‘estimated’

Reconcile singular/plural. ‘area’ -> ‘areas’

Delete ‘grey’ (triangles are now green)

Response: Changed. Added. Deleted.

Climate Change and Land Use Changes

AE: L141 – clarify what unit of observation was here. 1 data point per bee species (if so change wording at end to reflect this)? Or separate test for each species, in which case what was an observation, and how did you deal with spatial autocorrelation of climate pixels? (1st option seems more appropriate)

Response: For both the climate and land use changes, we conducted a single analysis per elevation profile. Therefore, the values included in the analyses are each 1 x 1km grid in each elevation profile where expected surveys were likely to have occurred. Therefore, the analysis is not species specific but represents the entire study area. We have clarified this, see lines 121-123:

“We extracted these metrics at a 1 × 1km grid resolution for a 10km buffer surrounding the centroid of all collection records. Therefore, the climate changes measured are representative of the entire study area and not only sites where species were collected.”

And lines 144-147:

“We extracted the land use within each elevation profile at a 1 × 1km grid resolution for a 10km buffer surrounding the centroid of all collection records within the same elevation profile. Therefore, the land use changes measured are representative of the entire study area and not only sites where species were collected.”

AE: L143 – acronyms make for hard reading. Recommend replacing ‘LULC’ with simply ‘land use’ & explain that this shorthand for Land use/Land cover

Response: Agreed, we have changed this throughout the manuscript. See line 135:

“Land use/land cover (here after referred to as land use)”

Editor Comments to Author: Abstract, line 35/36: 'provide preliminary evidence that some bumblebee species shift their ranges' (i.e. avoid the risk of 'some bumblebees shift' sounding like individuals, rather than the species). See also the AE's comments on the wording of this sentence.

Response: Thank you for pointing out this potential misinterpretation we have clarified that we mean species.

Reviewer(s)' Comments to Author:

Referee: 1

Comments to the Author(s).

R1: I have reviewed the revision of the manuscript “Bumblebees moving up: shifts in elevation ranges in the Pyrenees over 115 years”, and observed that the authors satisfactorily addressed most of my comments. I appreciate the new analysis of the shifts in climate ranges, and the fact that the potential effect of land-use/land-cover change is downplayed in the discussion, being largely hypothetical at this point.

Thanks also for the additional explanations about site selection and historical collections. Your replication strategy sounds more explicit to me. Still, I think you could be even more precise, in the sense that you should make clear in the text, in the appropriate “1889 Collections” section, what level of information you had regarding historical sampling locations. I understand broadly that this was only approximate, so that your replication was mainly about finding a place at similar altitude, in the vicinity of the supposed location, with the same plants if possible. However, I am still struggling a bit to understand if you knew, e.g. the name of a road along which plants/pollinators were sampled, or an indication like “after walking 1h from this place”, etc.

[Edit: after writing my comment, I searched online for MacLeod’s publication and, although I don’t speak Dutch, I was able to find how this information was recorded. It seems that for a given record you only have the name of the area, and an approximate altitude, e.g. P306 in MacLeod 1890: “*Bombus lapidarius*, Gèdre, 1200”. It was not entirely clear to me that the name of the 7 areas was the only location information you had! I understand now even better the reasoning behind your sampling strategy. If it wasn’t clear to me, maybe it could also be confusing for other readers.]

Response: We appreciate the comments on our improved manuscript and the newfound understanding our sampling methodology. We agree that this could be clearer. We have now made it explicitly clear that we only had access to area names and elevation from 1889 and that the 2005-06 sites were chosen based on this limited information.

See lines 164-167:

“He published an account of the plants and plant visitors he observed in 1891 in “De pyreneënbloemen”[20]. The names of these seven areas and the exact elevation at which he made observations was all the information published regarding his sampling locations. See Appendix A for a full description of each area sampled.”

And lines 179-182:

“In 2005 (8th to 25th August) and 2006 (14th to 31st August), two surveys were conducted to analyse the plant visitor community of the most visited plant species in 1889, in the same areas that MacLeod sampled. The sampling locations were estimated based on the seven sampling locations listed above and the exact elevations reported by Macleod for each plant species and insect visitor.”

Other minor comments:

R1: Fig 2: it may sound like a detail but I am definitely not a fan of p-values summarised by a star. Since you calculated it with precision, please write the p-value in full, otherwise it is unnecessarily hiding information.

Response: We have included the exact p-values alongside the stars. We have kept the stars as well, as we think this allows for quick and easy interpretation of the graph.

R1: L292: missing space between 15 and were

Response: Fixed

R1: L330-331: another study by the same group (Fourcade et al 2019, Biodiversity & Conservation) also discussed altitudinal shift in bumblebees in Scandinavia. You don't necessarily have to cite it but I mention it anyway, in case you had missed it.

Response: Indeed we had missed the publication of this article as we were in the process of writing the manuscript when it came out. The results are relevant to our study and we have referred to this in the discussion.

See lines 373-374:

"This contrasts with Fourcade et al.,[32.5] who observed elevation increases and species richness changes of the bumblebee community in a Norwegian sub-alpine habitat since the 1960s."

And lines 386-388:

"Fourcade et al.[33] found evidence that both climate and land use may have been responsible for changes to a sub-alpine bumblebee community but they too were unable to test this interaction directly."

R1: In conclusion, I believe this manuscript provides interesting results, relevant to both entomologists and climate change researchers. Anecdotally, I hiked this summer in the very same areas as described in this study and I couldn't help but think about it every time I encountered a bumblebee.

Response: We thank reviewer 1 for their detailed readings and improvements made to the manuscript and we are glad that our study added an extra dimension to visiting a beautiful location.

Referee: 2

Comments to the Author(s).

I think the author has responded to the reviewer's questions, made necessary corrections, and considerably enhanced the interpretation of this study. Adequate background information on the historical investigation and resurvey methodology has been provided so that the readers can evaluate the information and limitations of the findings. There is also better coherence in the analysis of the findings and their discussion.

Response: We thank reviewer 2 for their previous comments and detailed reading which has improved the manuscript.